# Peer review of "Adapting Short-Term Mentalization-Based Therapy to ICD-11 Personality Disorder in Adolescents"

_children, 2023, doi:10.3390/children10010093_

Round 1
Reviewer 1 Report
The authors have chosen a highly relevant topic. Early intervention for borderline personality disorder tailored to the patients' symptoms may reduce dysfunction and improve prognosis.
There were many strong points in the manuscript such as the developmental challenges such as identify formation that may occur in children or adolescents with borderline personality disorder and the detailed descriptions of MBT. There are several areas that could be enhanced.
The purpose for the manuscript was not clear at the beginning. In the abstract, a hypothesis is presented; however, there was no mention of a research design or methods. The authors later report that they have a study in progress (reference 16) that will address the hypothesis. The reader eventually understands that the manuscript contains a restricted review of chosen research/articles to describe MBT for youth with personality disorder. The manuscript could be strengthened with the use of a systematic approach, such as a systematic review of reviews or a systematic review of research on MBT -- that way important important research (e.g., https://www.mdpi.com/2077-0383/10/23/5622) might not be missed. The authors may also benefit from reviewing the research on stepped care for borderline personality disorder. Conversely, if the intent is to describe the MBT, then maybe the authors could develop a manual?
The authors stated that the ICD 11 "steps away from a medical model" which does not make sense since the the ICD 11 is an updated presentation of disorders. It is consistent with recent models used in medicine such as the biopsychosocial approach and the International Classification of Functioning Disabilty and Health.
This sentence in the manuscript is also unclear: "Understanding personality disorder severity through the lens of mental-90 ization has important implications, as the manifestations of the disorders are developmen-91 tal in origin, dynamic and process-based rather than a reflection of static traits inherent in the individual" since the manuscript is focused on personality disorder - "an enduring pattern of inner experience and behavior" (DSM 5).
On page 5, the authors noted the strengths of having one of the group therapists double as the indiviudal therapist; the potential disadvantages should also be noted such as the youth divulging information in the individual session that the youth would not like known in the group session, and the fear this potential issue may illicit.
Reviewer 2 Report
The article goals / aim / hypothesis should be more clearly presented / summarized in the article.
In the abstract it is mention that:
(1) we propose that the short term mentalization-based therapy (MBT) program, ...., has potential clinical advantages for adolescents with ICD-11 personality disorder at the mild to moderate severity level.
(2) We hypothesize that this format, as opposed to the traditional long-term MBT format, can be beneficial for adolescents.
The two ideas could also be supported through a quantitative analysis of the researches (not just descriptive, the study si more descriptive in nature).
Round 2
Reviewer 1 Report
The authors have made some improvements to the manuscript; however, some concerns are still present.
The authors are now proposing a conceptual paper with no hypotheses, and this intent aligns better with the manuscript as written. The purpose is still unclear.
If the purpopse is to "make a case for the use of this format, as opposed to the traditional long-term MBT format", then the design should be a randomized clinical trial to test this research question.
It might be helpful for the authors to review the full defintion of Personality Disorder in the ICD 11 " Personality refers to an individual’s characteristic way of behaving, experiencing life, and of perceiving and interpreting themselves, other people, events, and situations. Personality Disorder is a marked disturbance in personality functioning, which is nearly always associated with considerable personal and social disruption. The central manifestations of Personality Disorder are impairments in functioning of aspects of the self (e.g., identity, self-worth, capacity for self-direction) and/or problems in interpersonal functioning (e.g., developing and maintaining close and mutually satisfying relationships, understanding others’ perspectives, managing conflict in relationships). Impairments in self-functioning and/or interpersonal functioning are manifested in maladaptive (e.g., inflexible or poorly regulated) patterns of cognition, emotional experience, emotional expression, and behaviour." As can be seen, the definition is broader than the authors contention that the focus is on function only. It is a personality disorder.
On page 7, the authors noted that they "will, thus, primarily review the evidence base for MBT in children and adolescence with borderline personality disorder...". In order to review the evidence base, a systematic review of effectiveness research should be conducted.
The idea of a conceptual paper instead of a report of research is fine but the stated purposes should align with the unsystematic approach taken (not a systematic study). Audiences may benefit from a description of, for example, the developmental aspects that should be considered when attempting to apply the long-term MBT and how MBT can be modified to align with the biopsychosocial development taking place during adolescence.
Author Response
Reviewer # 1
he authors have made some improvements to the manuscript; however, some concerns are still present.
Response: We thank the reviewer for acknowledging improvements and hope concerns are now sufficiently addressed.
The authors are now proposing a conceptual paper with no hypotheses, and this intent aligns better with the manuscript as written. The purpose is still unclear.
If the purpopse is to "make a case for the use of this format, as opposed to the traditional long-term MBT format", then the design should be a randomized clinical trial to test this research question.
Response: We completely agree with the reviewer but also feel this is exactly what we state on p. 16: “A recent systematic review assessing the effects of psychotherapies for adolescents with BPD [17] concluded that it is currently difficult to conclude whether psychotherapy in general is effective for adolescent with BPD, due to high risk of bias, high attrition rates, and small trials underpowered to confirm or reject realistic intervention effects on patient important outcomes. Therefore, there is a need for more high-quality trials with larger samples to identify potentially effective psychological interventions for adolescents with BPD. As we have outlined in this paper, the short-term MBT program can be a promising intervention for adolescents with ICD-11 personality disorder. To test this assumption, we suggest the next step to be a large randomized clinical trial randomizing adolescents to an adapted short-term MBT program as outlined in this paper or a control intervention using a psychological sham intervention.”
It might be helpful for the authors to review the full defintion of Personality Disorder in the ICD 11 " Personality refers to an individual’s characteristic way of behaving, experiencing life, and of perceiving and interpreting themselves, other people, events, and situations. Personality Disorder is a marked disturbance in personality functioning, which is nearly always associated with considerable personal and social disruption. The central manifestations of Personality Disorder are impairments in functioning of aspects of the self (e.g., identity, self-worth, capacity for self-direction) and/or problems in interpersonal functioning (e.g., developing and maintaining close and mutually satisfying relationships, understanding others’ perspectives, managing conflict in relationships). Impairments in self-functioning and/or interpersonal functioning are manifested in maladaptive (e.g., inflexible or poorly regulated) patterns of cognition, emotional experience, emotional expression, and behaviour." As can be seen, the definition is broader than the authors contention that the focus is on function only. It is a personality disorder.
Response: We acknowledge that this conceptual paper does not cover all aspects of personality as the point of this paper is mainly to provide new conceptual ideas for intervention as we are standing at the crossroads between ICD-10 and ICD-11. However, we have now added the following sentence: “In terms of the ICD-11 personality disorder classification, we will not cover all diagnostic requirements but focus on severity/functioning determinants.”
On page 7, the authors noted that they "will, thus, primarily review the evidence base for MBT in children and adolescence with borderline personality disorder...". In order to review the evidence base, a systematic review of effectiveness research should be conducted.
Response: From a general perspective we completely agree with the reviewer and because of this we start by citing the main conclusion from a systematic review from 2021: Jørgensen M.S.; Storebø O.J.; Stoffers-Winterling J.M.; Faltinsen E.; Todorovac A.; Simonsen E. Psychological therapies for adolescents with borderline personality disorder (BPD) or BPD features—A systematic review of randomized clinical trials with meta-analysis and Trial Sequential Analysis. PloS one. 2021, 16, 0245331; DOI: 10.1371/journal.pone.0245331.
The idea of a conceptual paper instead of a report of research is fine but the stated purposes should align with the unsystematic approach taken (not a systematic study).
Response: we agree and have now included the following sentence: “We acknowledge that a major limitation of the ideas presented in this conceptual paper is that the empirical literature was not systematically reviewed. The reader should take this into account when assessing potential benefits and harms of the ideas presented.”
Audiences may benefit from a description of, for example, the developmental aspects that should be considered when attempting to apply the long-term MBT and how MBT can be modified to align with the biopsychosocial development taking place during adolescence.
Response: We thank the reviewer for seeing these potential benefits.

Reviewer 2 Report
The authors could propose / mention possible public health policies, practical implications / solutions related to the analyzed topic(s).
Author Response
Reviewer # 2
The authors could propose / mention possible public health policies, practical implications / solutions related to the analyzed topic(s).
Response: We thank the reviewer for this suggestion and have included the following sentence: “Developing efficient short-term treatment for a diagnosis, which traditionally has been suggested to require treatment of longer duration could potentially have a large impact on public health policies and guidelines.”